evolution, ecology

host–pathogen coevolution, population size, population bottleneck, genetic drift, red queen dynamics

**Authors for correspondence:**
Andrei Papkou
e-mail: andrei.papkou@uzh.ch
Hinrich Schulenburg
e-mail: hschulenburg@zoologie.uni-kiel.de

# Population size impacts host–pathogen coevolution

Andrei Papkou[1,2], Rebecca Schalkowski[1], Mike-Christoph Barg[1], Svenja Koepper[1] and Hinrich Schulenburg[1,3]

[1]Department of Evolutionary Ecology and Genetics, Zoological Institute, Christian-Albrechts-Universitaet Kiel, 24098 Kiel, Germany
[2]Department of Evolutionary Biology and Environmental Studies, University of Zurich, Winterthurerstrasse 190, CH-8057 Zurich, Switzerland
[3]Max-Planck Institute for Evolutionary Biology, 24306 Plön, Germany

AP, 0000-0003-2104-5964; HS, 0000-0002-1413-913X

Ongoing host–pathogen interactions are characterized by rapid coevolutionary changes forcing species to continuously adapt to each other. The interacting species are often defined by finite population sizes. In theory, finite population size limits genetic diversity and compromises the efficiency of selection owing to genetic drift, in turn constraining any rapid coevolutionary responses. To date, however, experimental evidence for such constraints is scarce. The aim of our study was to assess to what extent population size influences the dynamics of host–pathogen coevolution. We used *Caenorhabditus elegans* and its pathogen *Bacillus thuringiensis* as a model for experimental coevolution in small and large host populations, as well as in host populations which were periodically forced through a bottleneck. By carefully controlling host population size for 23 host generations, we found that host adaptation was constrained in small populations and to a lesser extent in the bottlenecked populations. As a result, coevolution in large and small populations gave rise to different selection dynamics and produced different patterns of host–pathogen genotype-by-genotype interactions. Our results demonstrate a major influence of host population size on the ability of the antagonists to co-adapt to each other, thereby shaping the dynamics of antagonistic coevolution.

## 1. Introduction

The evolutionary success of species depends on their ability to adapt to a changing environment. Adaptation to other coexisting species can be particularly challenging because these species themselves are subject to ecological and evolutionary changes, yielding highly variable selection pressures. Host–pathogen interactions are an example of these dynamics, and if ongoing, they can trigger a process of coevolution, i.e. a series of adaptations and counter-adaptations between the interacting species [1]. The failure to rapidly adapt to a coevolving antagonist can have devastating consequences for a population [2,3]. Therefore, coevolution favours traits that can accelerate adaptation, such as increased mutation and recombination rates [4–6], sexual reproduction and horizontal gene transfer [7–10], high standing genetic diversity in populations and the expansion of virulence/immunity gene families [11–13].

Population size is a key characteristic defining the ability of populations to promptly respond to selection. It scales the input of novel genetic variants (via mutation and recombination) and affects the maintenance of standing genetic diversity. Population size also determines the efficiency of natural selection relative to genetic drift. Consequently, large populations have a higher evolutionary potential than populations of finite size or subjected to bottlenecks, which are constrained by low genetic diversity and genetic drift [14]. Theoretical studies

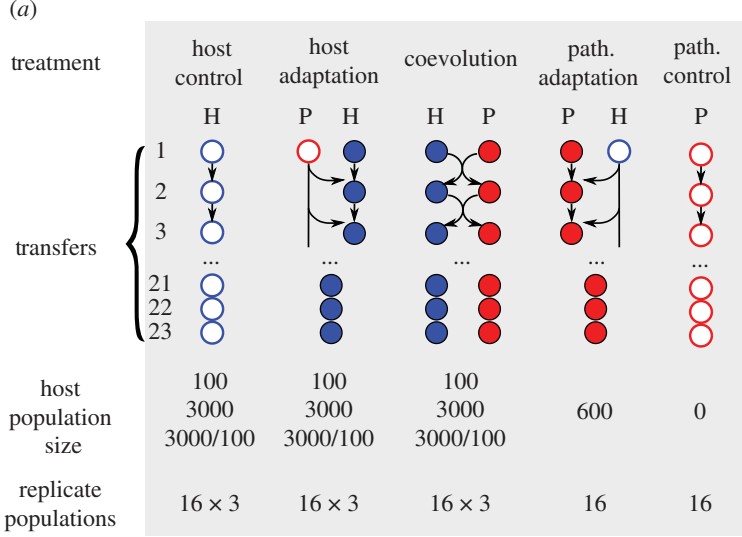

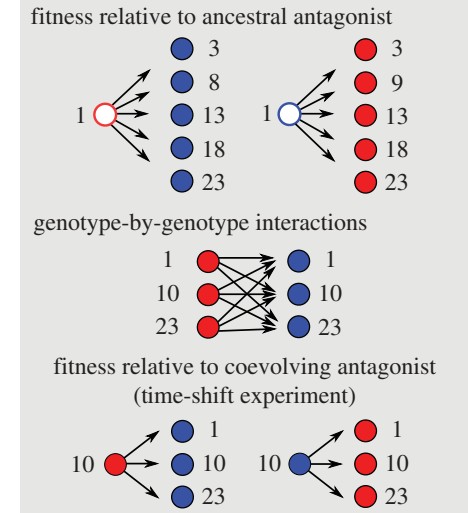

**Figure 1.** Design of the evolution experiment and subsequent characterizations. (*a*) The evolution experiment consisted of five main treatments and three population size manipulations. Host and pathogen were evolved either in the presence of a coevolving antagonist (middle treatment), a non-adapting antagonist taken from a stock culture (second and fourth treatments), or in the absence of an antagonist (far left and far right treatments). The treatments with evolving hosts were repeated at a host population size of 100 (small populations), 3000 (large populations) or 3000 with periodic bottlenecks of 100 individuals at every fifth transfer (bottlenecked populations). We included 16 biological replicates for all treatment combinations. (*b*) Overview of subsequent characterizations. Evolved hosts and pathogens from transfers 3, 8, 9, 13, 18 and 23 were exposed to the respective ancestral antagonist (top illustration), allowing a comparison across the five main evolution treatments. We further performed time-shift experiments for the coevolution treatment, where hosts and pathogens from transfers 1, 10 and 23 were exposed to the co-adapting antagonist in all possible combinations (middle panel) or with a specific focus on the coevolved antagonist from transfer 10, in order to reconstruct the dynamics of coevolution. (Online version in colour.)

demonstrated that population size can change the rate of reciprocal adaptation [15], shift the outcome of antagonistic interactions in favour of the antagonist with a larger population size [16] and produce qualitatively distinct coevolutionary dynamics [17–19]. However, empirical evidence for these effects is scarce.

The approach of experimental evolution proved highly informative to study coevolution [20], but it has only rarely been used to assess the consequences of population size variation in this context [21–23]. Previous coevolution experiments with the red flower beetle and its microsporidian pathogen showed that in small host populations genetic drift had a stronger effect on genetic diversity [21] and recombination rate [22] than pathogen mediated selection. Furthermore, two independent bacteria-bacteriophage coevolution experiments showed that bottlenecks in bacterial populations led to rapid phage extinction [2,24]. Two non-exclusive alternatives may explain the results: (i) strong bottlenecks and thus drift removed rare sensitive bacteria, which served as phage reservoirs, from generally resistant host populations [2]; and (ii) phage infectivity was reduced by diluting phage titres during bottlenecks, similarly leading to phage extinction [24]. The importance of genetic drift on coevolution thus remains elusive in the latter studies. More generally, ecologically driven changes and epidemiological feedbacks influence the dynamics of coevolution both in nature [14] and in laboratory experiments [25–28], complicating the inference of drift effects. For instance, most bacteria-bacteriophage coevolution studies use the batch culturing method, where bacteria and phage densities change dynamically owing to multiple infection-burst-infection cycles [2,4,20,29]. This epidemiological process affects the probability of infection, relaxing or increasing selection at different time intervals of batch culturing, making it difficult to reconstruct the influence of population

size. In addition, so far, it is unclear whether periodic bottlenecks, which are common in nature [14], cause similar genetic drift effects on coevolution than those known for continuously small populations.

The aim of our study was to assess the effect of host population size and periodic bottlenecks on coevolution using the nematode *Caenorhabditis elegans* and the Gram-positive bacterium *Bacillus thuringiensis* as an established, laboratory-based host–pathogen model [30–33]. *Bacillus* spore-toxin mixtures infect *C. elegans* upon oral uptake, ultimately causing worm death [31,32,34]. This model was previously used to characterize the nematode's immune system [32,34] and the dynamics of host–pathogen coevolution, demonstrating rapid co-adaptation of the two antagonists owing to strong reciprocal selection [30,31,33], consistent with antagonistic frequency-dependent selection dynamics (aFDS; [33]), apparently driven by copy number variation of toxin-containing plasmids in the pathogen [30,33] and outcrossing in the host [8,31]. For this study, we developed a new coevolution protocol to vary census host population size, while keeping the density of hosts and pathogens constant. We tracked coevolutionary interactions across 23 host generations in small versus large populations and also large populations with periodic bottlenecks (figure 1). Because both *C. elegans* and *B. thuringiensis* survive cryo-preservation, we froze evolving material in regular intervals, thereby creating a 'fossil' record for subsequent reconstruction of the evolutionary dynamics. We specifically asked whether host population size (i) has a general effect on adaptation under coevolution and non-coevolution conditions (based on comparisons of evolved hosts or pathogens with their respective ancestral antagonists; figure 1*b*, top panel), and (ii) influences coevolutionary dynamics (based on time-shift experiments with evolved hosts and pathogens from the coevolution treatment; figure 1*b* middle and bottom panels).

# 2. Material and methods

## (a) Material

A genetically diverse *C. elegans* population was provided by H. Teotonio [35]. *Caenorhabditis elegans* was maintained in V-medium [33] with the addition of *Escherichia coli* OP50 as a food source ($5 \times 10^9$ cell ml$^{-1}$). The V-medium does not support growth of *E. coli* or *B. thuringiensis*. Prior to coevolution, *C. elegans* was pre-adapted to the medium at 16°C in 16 replicate populations, initiated with 3000 L1-stage larvae in 3000 µl microcosms and transferred once per week for a total of 22 weeks (equivalent to 22 host generations). After the last transfer, all 16 populations were combined and frozen at −80°C.

The nematocidal *B. thuringiensis* strain MYBT18247 was used as a pathogen. It was transformed using plasmid pHT315 provided by C. Nielsen-LeRoux [36], expressing red or green fluorescent protein (RFP or GFP). Infectious spores were produced by incubating *B. thuringiensis* in T3-medium for 4 days at 25°C [33].

## (b) Coevolution experiment

The methods for evolving the large host population size have been published previously [33]. The treatments for small and also for the bottlenecked host population size are unpublished, yet were run in parallel to the previously published experiment.

### (i) Experimental design

Pre-adapted *C. elegans* were allocated to three experimental treatments: coevolution, host adaptation and host control (figure 1*a*). During coevolution, hosts were transferred together with coevolving pathogens for 23 host generations. During host adaptation, only *C. elegans* was transferred, and *B. thuringiensis* was taken from the original stock allowing evolution exclusively in the host. During host control, host populations only adapted to the experimental protocol without any pathogen. To test the effect of population size, each treatment included three types of *C. elegans* populations: small (*n* = 100 nematodes), large (*n* = 3000) and bottlenecked populations (*n* = 3000, reduced to 100 at every fifth transfer). To manipulate host population size, the use of different multi-well plates and microcosm volumes ensured identical surface-to-volume ratios. Large populations were maintained in 3000 µl microcosms using six-well plates and small populations in 100 µl using 96-well plates (in all cases, 1 worm µl$^{-1}$). Bottlenecked populations were kept in 3000 µl and reduced to 100 µl at every fifth transfer. A transfer following a bottleneck was performed in 600 µl (24-well plates) to maintain worm density close to 1 worm µl$^{-1}$ during population expansion. In total, the experimental design for the host consisted of nine combinations of treatments and population sizes and had 16 replicate populations per combination (3 treatments × 3 population size types × 16 replicates = 144 populations). Two more treatments were included for the pathogen with 16 replicates each: the pathogen adaptation treatment, in which hosts were taken from the stock allowing only the pathogen to evolve, and the pathogen control treatment, in which pathogens evolved without host (figure 1*a*). During the experiment, host and pathogen populations were regularly sampled and cryo-preserved at −80°C for subsequent characterization.

### (ii) Selection procedure

Experimental evolution was carried out in 7-day transfers [33]. On day 1, L1-instar *C. elegans* larvae were inoculated into fresh V-medium. The number of larvae was determined by counting them in a 5 µl drop under a dissecting scope, averaging three independent counts to calculate the inoculation volume for each population. On day 5, host populations (L4-instar larva stage) were infected by adding *B. thuringiensis* spores ($2 \times 10^8$ spores ml$^{-1}$). This spore dose causes high host mortality (mean ± s.d.: 0.819 ± 0.104 for the ancestral host and pathogen). However,

nematode eggs fertilized prior to parental death remain viable in the cadavers and may thus also contribute to the next host generation in our experimental set-up. At day 7, all (dead and alive) *C. elegans* were bleached in 1% sodium hypochlorite to kill bacteria and obtain eggs for the next transfer [33]. This mass selection protocol should favour high host reproduction.

To obtain *B. thuringiensis* for the next transfer, approximately 30 nematodes (dead or alive) were sampled from each population after 40 h of infection but before bleaching. These samples were maintained for 2 more days (until day 2 of the next transfer) to finalize the sporulation of pathogens growing inside the cadavers. The cadavers were pasteurized (10 min at 80°C) to kill *E. coli* and inoculated into T3-medium for the production of infectious spores. On day 5, the spores were harvested by centrifuging and normalized using optical density; $2 \times 10^9$ spores ml$^{-1}$ were mixed with *E. coli* in 1 : 10 proportion and used to infect host populations.

## (c) Fitness assays

All fitness assays were based on a repetition of one transfer of experimental evolution, in which host and pathogen from different generations were combined to measure phenotypic changes. The nematodes were recovered from −80°C and propagated for 1–2 generations to eliminate any effects from freezing. *Bacillus thuringiensis* was similarly revived from −80°C. Infection was initiated by exposing L4 larvae/young adults to infectious spores (either ancestral or coevolved spores; figure 1*b*), followed by phenotypic analysis after 40 h. For the large population size, the results on host fertility, pathogen competition and the time-shift experiment have been previously published (electronic supplementary material, table S1) [33]. All other results are unpublished.

### (i) Host fertility

Eight out of 16 host populations were randomly chosen from generations 3, 8, 13, 18, 23 and exposed to $2 \times 10^8$ spore ml$^{-1}$ of the ancestral *B. thuringiensis* (figure 1*b*, top panel). After 40 h, worms were fixed in 10 mM sodium azide and photographed with a Lecia M205-FA microscope. Up to 30 hermaphrodites were selected for each population following a pre-defined grid in each image. The mean number of eggs per hermaphrodite was used as a proxy for host fertility.

### (ii) Pathogen competition

All evolved strains expressed RFP, while the competitor reference strain expressed GFP. *Bacillus thuringiensis* from generations 0 and 23 were recovered from −80°C, mixed in 1 : 1 ratio and used to infect the ancestral host population at $2 \times 10^8$ spores ml$^{-1}$ in three replicates. After 40 h, nematodes were washed and crushed to release bacteria using a tissue homogenizer (1200 r.p.m., 3 min) and zirconia beads. Cell suspensions were serially diluted, and after 2 days at 25°C on T3 agar, green and red colonies were counted to estimate relative fitness of evolved strains using the formula: $s = [(R_1/G_1)/(R_0/G_0)] - 1$, where $R_0$, $R_1$, $G_0$ and $G_1$ are the numbers of red and green colonies before and after selection (see SI Appendix in [33]). This measure of fitness integrates a change in genotype frequencies over the whole round of infection neglecting how many cell divisions happened during the infection, which cannot be easily determined.

### (iii) Pathogen virulence

Pathogen populations from generations 0, 3, 9, 13, 19 and 23 were used to infect the ancestral worms at $2 \times 10^8$, $1 \times 10^8$ or $0.33 \times 10^8$ spores ml$^{-1}$. After 40 h, the proportion of dead worms served to estimate pathogen virulence in a sample of 30 hermaphrodites.

### (iv) Host resistance

Host populations from generations 0 and 23 were exposed for 40 h to the ancestral pathogen at $0.33 \times 10^8$ spores ml$^{-1}$. The proportion of dead hermaphrodites served to infer host resistance.

A lower spore dose was chosen to improve the sensitivity of the assay because host mortality increased in the evolved hosts.

### (v) Genotype × genotype interactions and time-shift experiments

Host and pathogen populations from transfers 1, 10 and 23 from the same replicate of the coevolution treatment (large, small and bottlenecked populations) were combined in all possible time-point combinations (figure 1b), resulting in 432 infection experiments (9 time-shifts × 3 population size treatments × 16 evolved replicate populations). Host populations were infected with matching pathogen lines using $10^8$ spore ml$^{-1}$. After 40 h, virulence and eggs per hermaphrodite were scored as above.

### (d) Statistical analysis

All statistical analyses were performed in R. Regression analysis served to assess host fertility, resistance and virulence data. Experimental treatments (control, adaptation, coevolution), population types (large, small or bottlenecked) and infection doses (whenever more than one dose was used) were modelled as fixed predictors, generation time as continuous predictor and population identities of biological replicates as random intercepts. Mixed-effect models were fitted with restricted maximum likelihood using the package lme4 v. 1.1-19 [37]. Post hoc comparisons were performed for significant main effects and p-values were adjusted using Tukey or the single-step methods (packages emmeans v. 1.3.2, multcomp v. 1.4-8) [38,39]. The data for pathogen virulence were not normally distributed and were transformed by Box-Cox power transformation (lambda parameter was chosen using package MASS v. 7.3-54 [40]). In addition, the pathogen virulence data did not change linearly over time, hence polynomial coefficients were introduced into the model to account for nonlinearity. Host fertility and host resistance data were fitted without transformation.

Pathogen fitness was analysed using arithmetic means of three technical measurements per population. Experimental treatments and population sizes were compared using two-sided Wilcoxon rank-sum tests and p-values adjusted by the Holm–Bonferroni method.

To analyse genotype × genotype (G × G) interactions, a statistical modelling framework was applied analogous to genotype × environment (G × E) interaction analysis [41]. Two host–pathogen pairs were excluded from the analysis, because one could not be recovered after freezing, while in the other case, the pathogen completely lost virulence. This resulted in 16 host–pathogen pairs for the small populations and 15 pairs for each the large and the bottlenecked populations. Linear-mixed models were fitted to estimate slope of phenotypic change for two time intervals (H1 to H10 and H10 to H23) and separately for each pathogen time-point (P1, P10 and P23). Post hoc tests were used to compare the estimated slope coefficients within the same time interval across pathogens as an indicator of G × G interactions. p-values were corrected for multiple testing using the single-step method (n = 27) [42].

To assess coevolutionary selection dynamics, we analysed time-point combinations with either H10 (i.e. P1-H10, P10-H10 and P23-H10) or, alternatively, P10 (P10-H1, P10-H10, P10-H23). We used the 16 biological replicates to determine frequencies of each of the four possible coevolutionary patterns (1 > 10 > 23, 1 < 10 < 23, 1 < 10 > 23, 1 > 10 < 23) and tested whether their distribution deviates from null expectation (0.25, 0.25, 0.25, 0.25), using exact multinomial tests (including Holm–Bonferroni adjustment of p-values).

## 3. Results

### (a) Change in fitness relative to ancestral antagonist upon coevolution and non-coevolution conditions

To assess if antagonistic selection led to a change in host fitness, we randomly selected eight out of 16 replicate host

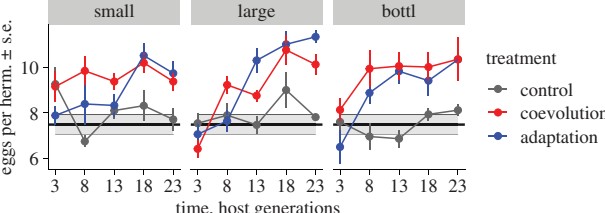

**Figure 2.** Evolutionary change in host fertility. Fertility was measured as average number of eggs per hermaphrodite after 40 h exposure to the ancestral pathogen. Different colours indicate the treatments (control treatment in grey, adaptation in blue and coevolution in red). Left, middle and right panels correspond to small, large and bottlenecked host populations. The error bars show standard error of means (s.e.) for eight replicates (eight lines, 30 hermaphrodites per line). The black line and grey shaded area show the mean ± 1 s.e. of the ancestral host population. (Online version in colour.)

populations from five time-points and infected them with the ancestral *B. thuringiensis* (figure 1b, top panel). We found that the average egg number per hermaphrodite gradually increased under coevolution and host adaptation but not control conditions (figure 2; electronic supplementary material, tables S2 and S3). The increase was observed for all population sizes including small populations. However, a significant interaction between time and population size suggests that population size influenced the rate of adaptation ($F_{2,304}$ = 6.930, p = 0.00114, electronic supplementary material, table S2). At the endpoint, large and small populations did not differ significantly (post hoc Wald test t = −2.0967, p = 0.25143). When we re-examined host fertility at the endpoint using more biological replicates and two different infection doses (electronic supplementary material, figure S1 and table S4), the coevolved hosts from large populations had consistently more eggs than those from small populations (electronic supplementary material, table S5), indicating that small population size constrained host adaptation. No significant differences were found between large and bottlenecked populations, or between bottlenecked and small populations, or between the coevolution and host-adaptation treatments.

Coevolution with a pathogen frequently promotes the evolution of resistance [31,43,44]. In our analysis, we found no increase but rather a decrease in resistance of evolved hosts towards ancestral *B. thuringiensis* (electronic supplementary material, figure S2, tables S6 and S7). Although at first sight unexpected, this result can be explained by the fact that host resistance was not under direct selection. During the experiment, C. elegans eggs were harvested from both alive and dead hermaphrodites, thus selecting the genotypes which would produce the most eggs and not necessarily highest resistance (the embryos inside eggs are protected from pathogens and remain viable after parent death). We further found no effect of population size on resistance during coevolution ($F_{2,131.194}$ = 1.749, p = 0.17805) and conclude that hosts adapted to pathogens by increasing fertility.

To assess pathogen adaptation, we measured relative competitive fitness of evolved versus ancestral *B. thuringiensis* in ancestral *C. elegans*. Although most treatments revealed increasing levels of pathogen fitness, only coevolved pathogens from large populations had significantly higher fitness than ancestral bacteria (electronic supplementary material, figure S3; Wilcoxon rank-sum test W = −47.49, p = 0.0158967), while no other comparison was significant (electronic

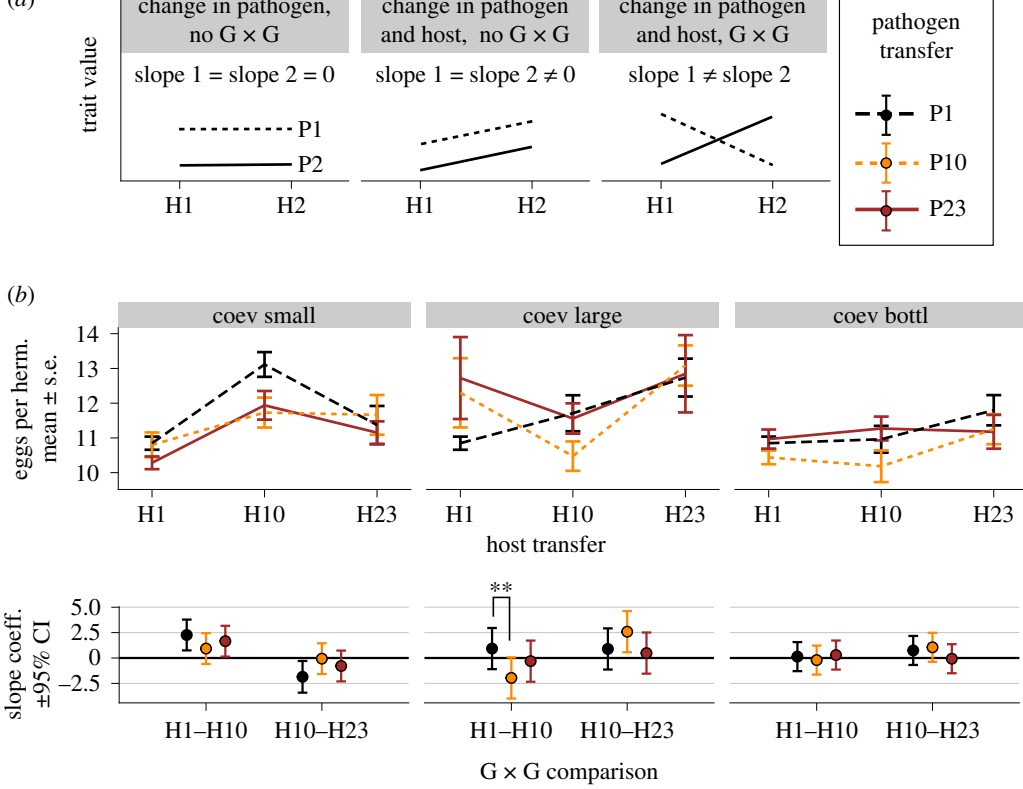

**Figure 3.** The emergence of host–pathogen $G_H \times G_P$ interactions during coevolution. (a) Hypothetical reaction norms for presence or absence of $G_H \times G_P$. A cross-infection experiment combining hosts (H1, H2) and pathogens (P1, P2) from two coevolution time-points may reveal three scenarios: (left panel) parallel horizontal lines indicating a trait change in one antagonist only; (middle panel) parallel non-horizontal lines indicating a change in both antagonists without $G_H \times G_P$; or (right panel) non-parallel lines suggesting $G_H \times G_P$ interactions. (b) Reaction norms for host fertility. The top panels show reaction norms reconstructed by measuring host fertility in all combinations of hosts and pathogens from transfers 1, 10, 23 (15–16 biological replicates per time-point combination; error bars show standard error of means). Different colours depict results for pathogens from 1, 10 and 23. Left, middle and right columns show results for small, large and bottlenecked populations, respectively. The bottom panels show slope coefficients estimated for the corresponding reaction norm lines from the top panels. The error bars indicate 95% confidence intervals adjusted using the single-step method. The absence of $G_H \times G_P$ would have similar slopes, while $G_H \times G_P$ interactions non-equal slopes. Statistically significant $G_H \times G_P$ were identified for interactions at transfers 1 and 10 in large populations (electronic supplementary material, tables S12 and S13). (Online version in colour.)

supplementary material, table S8). We further examined changes in virulence by scoring mortality in ancestral *C. elegans* (electronic supplementary material, figure S4). As with host resistance, the coevolution treatment did not directly select for virulence: pathogens were sampled from both dead and alive hosts, probably selecting genotypes with faster within-host replication. Unsurprisingly, we found a significant reduction in virulence across treatments (electronic supplementary material, table S9). In the coevolution and the pathogen adaptation treatments, the decrease in mean virulence was mostly caused by a virulence loss in some replicate populations, while other replicates maintained high virulence (electronic supplementary material, figure S4). By contrast, most replicates from the control treatment showed significantly reduced virulence (electronic supplementary material, table S10). These findings suggest that pathogens evolving with hosts maintained higher virulence than controls without host (post hoc Wald test for transfer 23, cont - coev large = −0.171, d.f. = 546.27, t = −3.718, p = 0.00206).

## (b) The effect of population size on $G_H \times G_P$ interactions during host–pathogen coevolution

The results presented thus far were obtained by exposing the evolved populations to their ancestral antagonists. However, during coevolution, the relevant fitness characteristics should

depend on concurrent changes in the co-adapting antagonists and may be shaped by genotype-by-genotype interactions ($G_H \times G_P$ analogous to G × E interactions; figures 1b and 3a) [41,45]. Therefore, we characterized $G_H \times G_P$ interactions within a coevolving replicate for host mortality and host fertility and all possible time-point combinations of hosts and pathogens from transfers 1, 10 and 23 (figure 1b, middle panel), including more than 400 infection assays and 20 000 hermaphrodites. For host mortality, the comparisons did not yield any significant differences (pathogen transfer × host change $F_{4,345.393}$ = 2.317, p = 0.05694, electronic supplementary material, table S11 and figure S5). By contrast, our analysis of host fertility (which was under direct selection during experimental coevolution) revealed significant $G_H \times G_P$ interactions for transfers 1 and 10 in the large populations (figure 3b; pathogen transfer × host change $F_{4,344.730}$ = 5.819, p = 0.00007; electronic supplementary material, table S12). Interestingly, the regression coefficients estimated using pathogens from transfers 1 and 10 had opposite signs suggesting that evolutionary changes in the host led to opposite fitness outcomes dependent on the pathogen context (figure 3b; electronic supplementary material, table S13). Thus, host coevolution resulted in higher fertility when measured in the presence of pathogens from transfer 1, and, simultaneously, lower fertility in the presence of pathogens from transfer 10 (figure 3b). In addition, we found that host fertility changes of large

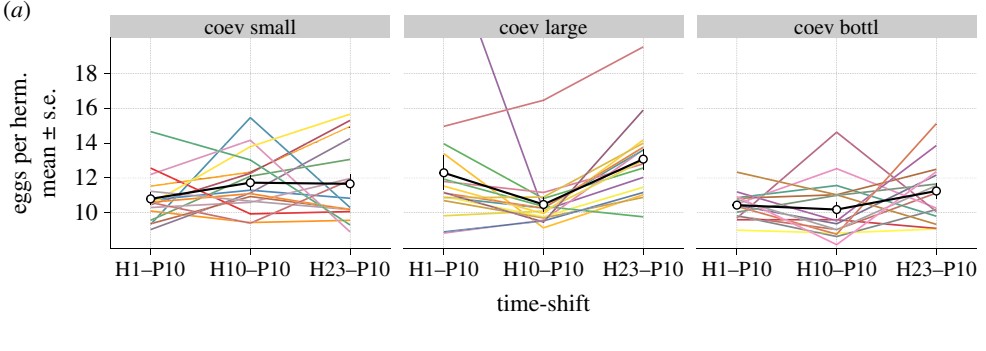

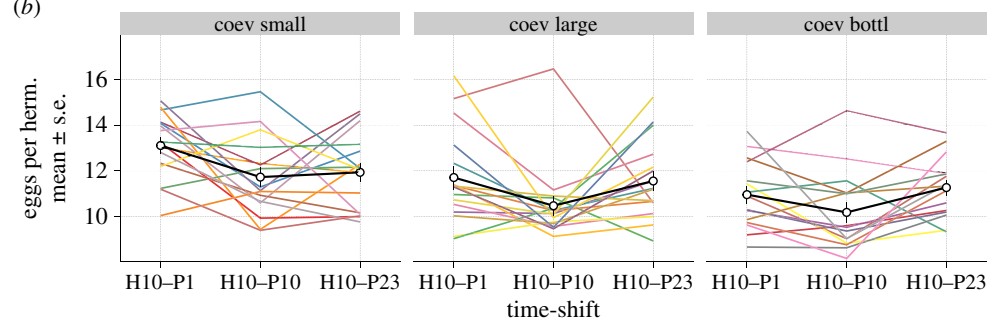

**Figure 4.** Dynamics of coevolution inferred from time-shift experiments for focal antagonists from transfer 10. (*a*) The time-shift experiment for focal pathogen from transfer 10 (P10) exposed to past, contemporaneous and future pathogens (from transfers 1, 10 and 23; H1, H10 and H23, respectively). The left, middle and right panels correspond to small, large and bottlenecked populations. The host and pathogen were always matched with the coevolved antagonist from the same replicate population. Each line represents an independent replicate; the black lines and bars show the average across replicates with standard errors (15–16 lines, 30 hermaphrodites per line). (*b*) Similar time-shift experiments using focal host from transfer 10 (H10). The presented data are a subset of that used in figure 3*b*. (Online version in colour.)

populations were significantly different from those of small populations (host change × pop size, $F_{4,346.464} = 12.765$, $p < 0.00001$; electronic supplementary material, table S14), indicating divergence of $G_H × G_P$ interaction patterns between different population sizes.

## (c) Coevolutionary dynamics

We characterized coevolutionary dynamics using time-shift experiments [46], in which one species is exposed to its antagonist from the evolutionary past, present and future (figure 1*b*, bottom panel). The two expected outcomes are: (i) higher fitness in populations from the future (past < present < future), indicating recurrent selective sweep dynamics; and (ii) higher fitness in a contemporaneous combination (past < present > future), consistent with aFDS (or Red Queen dynamics). We used part of the data from figure 3 for this analysis of coevolution patterns. Previously, we reported that the coevolved host–pathogens pairs from large populations showed time-shift patterns for host fertility which were most consistent with aFDS (also shown in the middle panel on figure 4*a,b*) [33]. In our additional analysis of host fertility for the small and bottlenecked populations, we only observed consistent aFDS patterns in the pathogen lines from bottlenecked populations (the right panel on figure 4*b*; electronic supplementary material, table S15), but not small populations. Our results suggest that small population size and, to lesser extent, bottlenecked populations produced selection dynamics different from those in large populations. Conversely, host mortality data from time-shift experiments did not reveal significant differences between population types (electronic supplementary material, figure S6 and table S16).

## 4. Discussion

In this study, we combined host–pathogen experimental evolution with time-shift experiments to assess the influence of host population size on host–pathogen coevolution. We found that population size generally increased fitness of the hosts from coevolution and host-adaptation conditions, when these were exposed to the ancestral pathogen (figure 2; electronic supplementary material, figure S1). More importantly, when coevolved hosts and pathogens were exposed to co-adapted antagonists from different time-points, host population size caused distinct $G_H × G_P$ interaction patterns (figure 3) and significantly affected coevolutionary dynamics (figure 4). While we previously reported that the large populations from this experiment showed phenotypic signatures consistent with aFDS in the coevolving antagonists [33], our results have now revealed that coevolution in small populations and, to lesser extent, in the bottlenecked populations is inconsistent with aFDS (figures 3 and 4).

The observed differences between large, small and bottlenecked populations can be attributed exclusively to evolutionary processes. In detail, we controlled host population size at every generation, and the host density per unit of volume was the same across treatments. In addition, each transfer of coevolution consisted of only a single round of infection, excluding the possibility of epidemiological feedbacks. Therefore, our results are unlikely to be affected by epidemiological or density-dependent effects. The reduced rate of adaptation in small host populations can thus be explained by (i) reduced selection efficiency, (ii) stochastic loss of beneficial alleles, both as a consequence of genetic drift, and/or (iii) a reduced effective rate of recombination and longer persistence of linkage disequilibrium, reducing the generation of new

favourable allele combinations. Our results are generally consistent with a recent study finding that small populations of *C. elegans* ($n = 50$) failed to evolve resistance when they were initially sensitive to a pathogen and to maintain resistance when initially resistant [47]. In this previous study, large populations ($n = 500$) succeeded in both evolving and maintaining resistance, suggesting that genetic drift negatively affected selection and maintenance of favourable alleles in small populations. Interestingly, in our study, the small populations consistently showed fitness improvements (coevolution and host-adaptation treatments, figure 2) and maintained high fitness until the end (electronic supplementary material, figure S1). Perhaps, the strength of genetic drift was not as overwhelming in our experiment (i.e. our small populations had a census size of $n = 100$) and/or the small populations possessed higher genetic variation than in the published study, thus allowing them to sustain adaptation.

The observed variation in evolutionary responses among population size treatments was unlikely owing to variation in initial genetic diversity. All initial host populations were established from the same highly genetically diverse *C. elegans* population with more than 370 000 single-nucleotide polymorphisms [33], in order to ensure that the treatments only varied in host population size and in coevolution with an antagonist, but not initial genetic diversity and initial genotypic composition, which can both influence the trajectory of coevolutionary adaptation [48,49]. Importantly, the high levels of initial genetic diversity used in our experiment are unlikely to be maintained in natural populations [50]. Therefore, we expect that natural populations of similar size would have a lower ability to adapt compared to the experimental populations. If the small populations had been allowed to reach an equilibrium level of genetic diversity before the start of the evolution experiment, they would probably not have shown such a consistent response to selection. Yet, the fact that, despite this high initial diversity, evolutionary adaptation was limited in the small populations shows that genetic drift had a considerable impact on adaptation within just a few generations. Similar observations were made in other previous coevolution [21–23] and mass selection experiments [37–39], in which adaptation was compared under different population sizes and with a non-equilibrated amount of genetic variation.

Host population size can also affect coevolution by influencing recombination and sexual reproduction. For example, in a previous study, a small population size in the red flour beetle resulted in higher recombination rates and changed mating behaviour during coevolution [22,23]. Recombination and sexual reproduction produce new allele combinations and thereby increase genetic diversity, which can help to rapidly adapt to coevolving pathogens [7,9,51]. *Caenorhabditus elegans* has an androdioecious mode of reproduction, such that hermaphrodites can reproduce sexually both by outcrossing with males or by selfing. We previously found

that the exposure to *B. thuringiensis* reduces outcrossing and increases selfing rates [8]. However, a lower level of outcrossing was stably maintained during coevolution potentially because it is sufficient to produce a beneficial level of genetic diversity and diverse offspring [8]. In future, it would be interesting to investigate how population size affects the rates of outcrossing, recombination and linkage discquilibrium in our system and how this affects coevolutionary dynamics at phenotypic and genomic levels.

Finally, host population size can impact coevolutionary dynamics owing to the intertwined nature of host–pathogen interactions [18]. For example, strong genetic drift can slow down host adaptation [47], and this in turn reduces selection intensity suffered by a pathogen [16]. Thus, we anticipated that the coevolution treatment will give rise to evolutionary trajectories that eventually will be different from trajectories in the adaptation treatment [30]. Surprisingly, our initial results in which we exposed evolved populations to the ancestral antagonists did not yield significant differences between these two treatments. However, this result was based on fitness measured using ancestral antagonists, which do not provide the relevant context for assessing fitness dynamics of coevolving species. When using co-adapted antagonist pairs in time-shift experiments, we found significant $G_H \times G_P$ interactions and identified the pattern of temporal co-adaptation. Moreover, these patterns were clearly different in large and small populations. We also detected a subtle but measurable effect of bottlenecks on coevolutionary dynamics, suggesting that the employed bottlenecking procedure did not have a strong impact on host adaptation. Taken together, our study demonstrates that small population size and to a lesser extent population bottlenecks constrain the dynamics of host–pathogen coevolution.

Data accessibility. All data are available in the electronic supplementary material [52]. All data and code to reproduce figures and analysis are also available from Figshare: https://doi.org/10.6084/m9.figshare.c.5656303.v1.

Authors' contributions. A.P.: conceptualization, formal analysis, investigation, methodology, validation, visualization, writing the original draft, writing the review and editing; R.S.: investigation, writing the review and editing; M.-C.B.: investigation, writing the review and editing; S.K.: investigation, writing the review and editing; H.S.: conceptualization, funding acquisition, project administration, resources, supervision, writing—original draft, writing the review and editing. All authors gave final approval for publication and agreed to be held accountable for the work performed therein.

Competing interests. The authors have no competing interests to declare.

Funding. This work was supported by the German Science Foundation within the priority program SPP1399 on host-parasite coevolution (grant nos. SCHU 1415/8 and SCHU 1415/9 to H.S.). H.S. has also received financial support from Germany's Excellence Strategy (grant no. EXC 2167-390884018, H.S.) and the Max-Planck Society (Max-Planck fellowship to H.S.).

Acknowledgements. We thank Henrique Teotónio, Christina Nielsen-LeRoux, Joachim Kurtz, Rebecca Schulte-Iserlohe and the Schulenburg laboratory for helpful advice.

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
