## [Peer Review File · Proceedings of the Royal Society B: Biological Sciences]

Review History

RSPB-2021-0607.R0 (Original submission)

Review form: Reviewer 1

Recommendation

Major revision is needed (please make suggestions in comments)

Scientific importance: Is the manuscript an original and important contribution to its field?

Good

General interest: Is the paper of sufficient general interest?

Good

Quality of the paper: Is the overall quality of the paper suitable?

Acceptable

Is the length of the paper justified?

Yes

Should the paper be seen by a specialist statistical reviewer?

No

Do you have any concerns about statistical analyses in this paper? If so, please specify them explicitly in your report.

No

It is a condition of publication that authors make their supporting data, code and materials available - either as supplementary material or hosted in an external repository. Please rate, if applicable, the supporting data on the following criteria.

Is it accessible?

Yes

Is it clear?

Yes

Is it adequate?

Yes

Do you have any ethical concerns with this paper?

No

Comments to the Author

This manuscript reports on the outcome of an experimental evolution study with the main treatment factor being population of small and big size. The study is well conducted and mostly clear to follow. The study design is related to that of earlier publications of the group, and is a success story of the lab. While I have no problems with the experiment, its analysis and presentation, but I have problems with the interpretation of the study design and therefore the interpretation of the results and the generalisations made.

The authors are surprised that "... small populations were still able to respond to selection, even if they comprised no more than 100 individuals.". I am not surprised! 23 generations is not a lot. A pop. size of 100 on the other hand is rather big. Given that the response to selection here is only attributable to standing genetic variation (mutations would not play a role with this exp. design), selection simply pulled out the fitter variants that were already at a reasonable frequency. Furthermore, the study suffers from an important misunderstanding of population genetics, which limits what can be concluded from simple theory. A consequence of drift is that small populations have lower genetic diversity than big populations. If a sub-population is created from a big population, it will have initially a too high genetic diversity. Only after several generations burn-in, a new equilibrium will be reached (This is why population geneticists use genetic diversity to estimate (effective) population size). In the current experiment, all populations had been recruited from a big stock. Thus, the genetic diversity, was over proportional high in the small populations. In fact, it was the same in all treatments! Thus, a relatively strong response to selection based on the standing genetic diversity is expected. While I do not consider this fatal for the article, this needs to be discussed and the generalisations made need to be adjusted accordingly. It also needs to be part of the discussion. This has also implications for the bottlenecked populations.

Another issue related to the founding population is the way the founding population was produced. The authors state that: "After the last transfer, all 16 populations were combined, mixed with DMSO (final concentration 15% v/v) and frozen at -80 °C. The pre-adapted population was designated as the ancestral population and used as a starting material for coevolution." As a consequence of this one-time mixing procedure, the populations exhibit strong linkage disequilibrium (LD), which will decline with time (recombination events) since mixing. LD can have profound impact on the response to selection, in particular when population size is small and the traits under selection being polygenic. Again, this needs to be worked out and discussed.

The discussion is rather boring und hardly informative. After the summary in the first paragraph, we learn across the next two paragraphs why the current study is great and other studies had different types of problems. The following paragraphs are also rather slim in discussing the biology behind the study. I would rather like to read a discussion telling me about the biology of the results presented here, about the possible roles of mutation and standing genetic variation, about the role of the specific experimental design for the outcomes, about the different forms of coevolution that are often discussed, about the role of selfing versus outcrossing of the host, about the long term perspective (what would we expect if the experiment would be run for 1000 generations) and more Thus, I would not add to the discussion, I would replace the current discussion with a new one.

I suggest to make clear that this is an entirely artificial system, where neither the host lives in its typical habitat nor is the pathogen a natural pathogen of these worms. Still, it is a good model system for basic questions.

The analysis of the experiment is very complex.
(although, I would like if the material would be presented in a less complex form. Not everything presented is needed.).

I suggest to use a more standardized design for the figures, e.g. use the same error bars (currently it is standard deviations and 95& C.I.).

Review form: Reviewer 2

Recommendation

Major revision is needed (please make suggestions in comments)

Scientific importance: Is the manuscript an original and important contribution to its field?

Good

General interest: Is the paper of sufficient general interest?

Good

Quality of the paper: Is the overall quality of the paper suitable?

Acceptable

Is the length of the paper justified?

Yes

Should the paper be seen by a specialist statistical reviewer?

No

Do you have any concerns about statistical analyses in this paper? If so, please specify them explicitly in your report.

Yes

It is a condition of publication that authors make their supporting data, code and materials available - either as supplementary material or hosted in an external repository. Please rate, if applicable, the supporting data on the following criteria.

Is it accessible?

Yes

Is it clear?

Yes

Is it adequate?

Yes

Do you have any ethical concerns with this paper?

No

Comments to the Author

This is a very interesting and elegant study investigating the effect of host population size on antagonistic coevolution between *C. elegans* and *B. thuringiensis*. I greatly appreciate the amount of work involved in collecting such an impressive data set!

My main concern is with the presentation of the work. For example, the authors refer to another paper to explain the evolution experiment. I would prefer a brief summary of what has been done in the main text.

Second, because of the scale of the study, a lot of information is hidden in the supplementary material. I found it quite hard to follow and interpret the main findings. The majority of statistical analyses (which are all sound) are described in the footnotes in the supplementary tables. Personally, I prefer a statistical summary of the main effects/interactions in the main body of the text, which might provide the reader with a bit more guidance on what results are discussed when.

Finally, I believe the authors need to be more explicit about the novelty of their work – do population bottlenecks and small sizes in theory differentially affect coevolutionary dynamics? The effect of host population bottlenecks on coevolution has been tested experimentally in bacteria-phage systems. From the introduction it is not immediately clear what this study might add to our understanding of how population size affects coevolutionary dynamics.

Abstract

Please clarify that population size of the host was manipulated and not that of the pathogen.

Introduction

Lines 53-62: Can the authors please explain how the effect of host population bottlenecks on antagonistic coevolution is expected to differ from that of small population size? This would clarify the novelty of this study (i.e. the effect of population bottlenecks on coevolution has been investigated experimentally, largely in bacteria-phage systems (e.g. Common & Westra 2019 and Hesse & Buckling 2016). How does this work add to our understanding?

Lines 75-77: Could they authors add a bit of background on the biology of the system. Does *B. thuringiensis* impose strong selection on *C. elegans*? Does it result in death or mainly reduced fecundity? Is *B. thuringiensis* ahead in the arms race (or is frequency dependent selection driving coevolutionary dynamics) and how is this likely to affect the host when its population sizes are reduced? I would like to know a bit more about this cool model system and why host population size could be a key determinant in driving coevolutionary dynamics.

Results

In general, there are no statistics provided in the result section. While I understand that not all contrasts can be given in the main text (as there are many), personally I would like to see data on significance of main treatment effects (and their interactions) without having to flick back and

forth between the main text and the supplementary material.

Lines 117-122: What I find perhaps more surprising is that adapted and coevolved host lines do not differ in mean egg number across all population sizes tested. This is not mentioned at all in results or in the discussion.

Lines 126-127: It is not clear from method section that 8 replicates were tested initially (it only says so in figure legend). Please clarify this, and how changing replicate numbers improved power of statistical models.

Line 129: There were no differences between small and bottlenecked populations either. Please mention this.

Lines 135: This is why I would like to see some more detail about the experimental procedure. It is not clear from the method section that eggs from all nematodes were collected and transferred and how this affected survival of bacteria (I presume the authors used this method because it is difficult to distinguish live/dead nematodes?). Is it known whether egg number varies as a function of nematode health (did dead individuals contribute less to future generations in your set up?)

Lines 138-140: This is not surprising given that main effect of population size was not significant (Table S6). I would omit this table. From my understanding, it is recommended to carry out post hoc comparisons only if main effect/interaction is significant.

Lines 141-146: This table provides information on contrasts, but there are no statistics on main effects of linear mixed models.

Line 274: should be "middle panel".

Lines 202-206: I think this information should be provided in the introduction when discussing coevolutionary dynamics. Is effect of host population size expected to be different for these different types of dynamics?

Discussion

I miss a discussion on why adapted and coevolved host populations did not differ in their final egg numbers (transfer 23) and how this links to observed FDS in large and bottlenecked populations (although less so).

Methods

Lines 305-327: The authors cite a related paper (Papkou et al 2019) to explain the experimental design. As a result, it is quite hard to understand the experimental evolution experiment (How do transfers work? How do you separate partners in evolution/adaptation treatment? How were population sizes remained constant within a season? (hosts have a generation time of one week)? I would like to see a brief summary of the actual experiment. I do realise that space is limited but perhaps this could be solved by placing the method section directly after the introduction. This way, the method summary at the beginning of the result section is superfluous and can be removed to free up some space.

Line 341: Are the exact sample sizes given somewhere (or are they always 30 per population)? How much variation is there within a population? By averaging, this information is lost (could include this in lme?).

Line 338: It is not immediately clear here whether these are ancestral spores or whether this is a general procedure used for all different host-parasite combinations. I like the brief summary

given in lines 110-116 (including reference to panel 1b), which could be placed here to clarify this?

Line 349: Please provide formula for relative fitness (it would seem from graph it is actually the selection coefficient).

Lines 361-362: Please specify the structure of the random effect as well as the explanatory variables and link function used (I presume a binomial error structure was used in case of survival data, but this is not provided in main text?). How were different model fits compared? Where assumptions of models met/tested (survival data are often skewed/zero-inflated)? This is not clear at all from statistics description and only becomes apparent from footnotes in the supplementary tables.

Lines 368-369: I presume post hoc comparisons were only performed when main effects/interactions were significant (see above for model comparison)?

Figure 1: "anatagoist" in panel B should be "antagonist"

Decision letter (RSPB-2021-0607.R0)

26-Apr-2021

Dear Dr Schulenburg:

I am writing to inform you that your manuscript RSPB-2021-0607 entitled "Population size impacts host-pathogen coevolution" has, in its current form, been rejected for publication in Proceedings B.

This action has been taken on the advice of referees, who have recommended that substantial revisions are necessary. With this in mind we would be happy to consider a resubmission, provided the comments of the referees are fully addressed. However please note that this is not a provisional acceptance.

To upload a resubmitted manuscript, log into <http://mc.manuscriptcentral.com/prsb> and enter your Author Centre, where you will find your manuscript title listed under "Manuscripts with

Decisions." Under "Actions," click on "Create a Resubmission." Please be sure to indicate in your cover letter that it is a resubmission, and supply the previous reference number.

Sincerely,
 Professor Hans Heesterbeek
 mailto: proceedingsb@royalsociety.org

Associate Editor
 Board Member: 1
 Comments to Author:

In this paper the authors use their *C. elegans*-*B. thuringiensis* system to investigate the impact of host population size on host-pathogen coevolutionary dynamics. I think this is an important topic that has received some limited theoretical attention in recent years but is still underexplored. Both reviewers found the paper interesting, but they also identified some shortcomings and made a number of insightful recommendations on how the paper can be improved.

Reviewer 1 raises important points about the way the initial populations were generated. I agree with the reviewer that this is a subtle but potentially important feature of the experimental design that needs to be clearly spelled out and discussed. How would the authors expect a different procedure for setting up the small populations to affect the results? (E.g. with a burn-in without any parasites to reduce genetic diversity to some level closer to what is expected in populations of this size.)

I also largely agree with reviewer 1's comment about the Discussion - it reads a bit like a cover letter and would benefit from a much more in-depth discussion of the results of the experiment, including the issue raised by Reviewer 2.

Finally, I agree with Reviewer 2 that the methods need to be described more exhaustively. Their suggestion to move the Methods section in front of the Results section would help reducing redundancies and is in line with the usual order of sections in Proc B papers.

Reviewer(s)' Comments to Author:
 Referee: 1
 Comments to the Author(s)

This manuscript reports on the outcome of an experimental evolution study with the main treatment factor being population of small and big size. The study is well conducted and mostly clear to follow. The study design is related to that of earlier publications of the group, and is a success story of the lab. While I have no problems with the experiment, its analysis and presentation, but I have problems with the interpretation of the study design and therefore the interpretation of the results and the generalisations made.

The authors are surprised that "... small populations were still able to respond to selection, even if they comprised no more than 100 individuals.". I am not surprised! 23 generations is not a lot. A pop. size of 100 on the other hand is rather big. Given that the response to selection here is only attributable to standing genetic variation (mutations would not play a role with this exp. design), selection simply pulled out the fitter variants that were already at a reasonable frequency. Furthermore, the study suffers from an important misunderstanding of population genetics, which limits what can be concluded from simple theory. A consequence of drift is that small populations have lower genetic diversity than big populations. If a sub-population is created from a big population, it will have initially a too high genetic diversity. Only after several generations burn-in, a new equilibrium will be reached (This is why population geneticists use genetic diversity to estimate (effective) population size). In the current experiment, all populations had been recruited from a big stock. Thus, the genetic diversity, was over proportional high in the small populations. In fact, it was the same in all treatments! Thus, a relatively strong response to selection based on the standing genetic diversity is expected. While I

do not consider this fatal for the article, this needs to be discussed and the generalisations made need to be adjusted accordingly. It also needs to be part of the discussion. This has also implications for the bottlenecked populations.

Another issue related to the founding population is the way the founding population was produced. The authors state that: "After the last transfer, all 16 populations were combined, mixed with DMSO (final concentration 15% v/v) and frozen at -80 °C. The pre-adapted population was designated as the ancestral population and used as a starting material for coevolution." As a consequence of this one-time mixing procedure, the populations exhibit strong linkage disequilibrium (LD), which will decline with time (recombination events) since mixing. LD can have profound impact on the response to selection, in particular when population size is small and the traits under selection being polygenic. Again, this needs to be worked out and discussed.

The discussion is rather boring und hardly informative. After the summary in the first paragraph, we learn across the next two paragraphs why the current study is great and other studies had different types of problems. The following paragraphs are also rather slim in discussing the biology behind the study. I would rather like to read a discussion telling me about the biology of the results presented here, about the possible roles of mutation and standing genetic variation, about the role of the specific experimental design for the outcomes, about the different forms of coevolution that are often discussed, about the role of selfing versus outcrossing of the host, about the long term perspective (what would we expect if the experiment would be run for 1000 generations) and more Thus, I would not add to the discussion, I would replace the current discussion with a new one.

I suggest to make clear that this is an entirely artificial system, where neither the host lives in its typical habitat nor is the pathogen a natural pathogen of these worms. Still, it is a good model system for basic questions.

The analysis of the experiment is very complex.
(although, I would like if the material would be presented in a less complex form. Not everything presented is needed.).

I suggest to use a more standardized design for the figures, e.g. use the same error bars (currently it is standard deviations and 95% C.I.).

Referee: 2

Comments to the Author(s)

This is a very interesting and elegant study investigating the effect of host population size on antagonistic coevolution between *C. elegans* and *B. thuringiensis*. I greatly appreciate the amount of work involved in collecting such an impressive data set!

My main concern is with the presentation of the work. For example, the authors refer to another paper to explain the evolution experiment. I would prefer a brief summary of what has been done in the main text.

Second, because of the scale of the study, a lot of information is hidden in the supplementary material. I found it quite hard to follow and interpret the main findings. The majority of statistical analyses (which are all sound) are described in the footnotes in the supplementary tables. Personally, I prefer a statistical summary of the main effects/interactions in the main body of the text, which might provide the reader with a bit more guidance on what results are discussed when.

Finally, I believe the authors need to be more explicit about the novelty of their work – do population bottlenecks and small sizes in theory differentially affect coevolutionary dynamics?

The effect of host population bottlenecks on coevolution has been tested experimentally in bacteria-phage systems. From the introduction it is not immediately clear what this study might add to our understanding of how population size affects coevolutionary dynamics.

Abstract

Please clarify that population size of the host was manipulated and not that of the pathogen.

Introduction

Lines 53-62: Can the authors please explain how the effect of host population bottlenecks on antagonistic coevolution is expected to differ from that of small population size? This would clarify the novelty of this study (i.e. the effect of population bottlenecks on coevolution has been investigated experimentally, largely in bacteria-phage systems (e.g. Common & Westra 2019 and Hesse & Buckling 2016)). How does this work add to our understanding?

Lines 75-77: Could they authors add a bit of background on the biology of the system. Does *B. thuringiensis* impose strong selection on *C. elegans*? Does it result in death or mainly reduced fecundity? Is *B. thuringiensis* ahead in the arms race (or is frequency dependent selection driving coevolutionary dynamics) and how is this likely to affect the host when its population sizes are reduced? I would like to know a bit more about this cool model system and why host population size could be a key determinant in driving coevolutionary dynamics.

Results

In general, there are no statistics provided in the result section. While I understand that not all contrasts can be given in the main text (as there are many), personally I would like to see data on significance of main treatment effects (and their interactions) without having to flick back and forth between the main text and the supplementary material.

Lines 117-122: What I find perhaps more surprising is that adapted and coevolved host lines do not differ in mean egg number across all population sizes tested. This is not mentioned at all in results or in the discussion.

Lines 126-127: It is not clear from method section that 8 replicates were tested initially (it only says so in figure legend). Please clarify this, and how changing replicate numbers improved power of statistical models.

Line 129: There were no differences between small and bottlenecked populations either. Please mention this.

Lines 135: This is why I would like to see some more detail about the experimental procedure. It is not clear from the method section that eggs from all nematodes were collected and transferred and how this affected survival of bacteria (I presume the authors used this method because it is difficult to distinguish live/dead nematodes?). Is it known whether egg number varies as a function of nematode health (did dead individuals contribute less to future generations in your set up?)

Lines 138-140: This is not surprising given that main effect of population size was not significant (Table S6). I would omit this table. From my understanding, it is recommended to carry out post hoc comparisons only if main effect/interaction is significant.

Lines 141-146: This table provides information on contrasts, but there are no statistics on main effects of linear mixed models.

Line 274: should be "middle panel".

Lines 202-206: I think this information should be provided in the introduction when discussing coevolutionary dynamics. Is effect of host population size expected to be different for these different types of dynamics?

Discussion

I miss a discussion on why adapted and coevolved host populations did not differ in their final egg numbers (transfer 23) and how this links to observed FDS in large and bottlenecked populations (although less so).

Methods

Lines 305-327: The authors cite a related paper (Papkou et al 2019) to explain the experimental design. As a result, it is quite hard to understand the experimental evolution experiment (How do transfers work? How do you separate partners in evolution/adaptation treatment? How were population sizes remained constant within a season? (hosts have a generation time of one week)? I would like to see a brief summary of the actual experiment. I do realise that space is limited but perhaps this could be solved by placing the method section directly after the introduction. This way, the method summary at the beginning of the result section is superfluous and can be removed to free up some space.

Line 341: Are the exact sample sizes given somewhere (or are they always 30 per population)? How much variation is there within a population? By averaging, this information is lost (could include this in lme?).

Line 338: It is not immediately clear here whether these are ancestral spores or whether this is a general procedure used for all different host-parasite combinations. I like the brief summary given in lines 110-116 (including reference to panel 1b), which could be placed here to clarify this?

Line 349: Please provide formula for relative fitness (it would seem from graph it is actually the selection coefficient).

Lines 361-362: Please specify the structure of the random effect as well as the explanatory variables and link function used (I presume a binomial error structure was used in case of survival data, but this is not provided in main text?). How were different model fits compared? Where assumptions of models met/tested (survival data are often skewed/zero-inflated)? This is not clear at all from statistics description and only becomes apparent from footnotes in the supplementary tables.

Lines 368-369: I presume post hoc comparisons were only performed when main effects/interactions were significant (see above for model comparison)?

Figure 1: "anatagoist" in panel B should be "antagonist"

Author's Response to Decision Letter for (RSPB-2021-0607.R0)

See Appendix A.

RSPB-2021-2269.R0

Review form: Reviewer 1

Recommendation

Accept as is

Scientific importance: Is the manuscript an original and important contribution to its field?

Acceptable

General interest: Is the paper of sufficient general interest?

Acceptable

Quality of the paper: Is the overall quality of the paper suitable?

Acceptable

Is the length of the paper justified?

Yes

Should the paper be seen by a specialist statistical reviewer?

No

Do you have any concerns about statistical analyses in this paper? If so, please specify them explicitly in your report.

No

It is a condition of publication that authors make their supporting data, code and materials available - either as supplementary material or hosted in an external repository. Please rate, if applicable, the supporting data on the following criteria.

Is it accessible?

Yes

Is it clear?

Yes

Is it adequate?

Yes

Do you have any ethical concerns with this paper?

No

Comments to the Author

none

Review form: Reviewer 2

Recommendation

Accept as is

Scientific importance: Is the manuscript an original and important contribution to its field?

Good

General interest: Is the paper of sufficient general interest?

Good

Quality of the paper: Is the overall quality of the paper suitable?

Excellent

Is the length of the paper justified?

Yes

Should the paper be seen by a specialist statistical reviewer?

No

Do you have any concerns about statistical analyses in this paper? If so, please specify them explicitly in your report.

No

It is a condition of publication that authors make their supporting data, code and materials available - either as supplementary material or hosted in an external repository. Please rate, if applicable, the supporting data on the following criteria.

Is it accessible?

Yes

Is it clear?

Yes

Is it adequate?

Yes

Do you have any ethical concerns with this paper?

No

Comments to the Author

I am happy with the changes the authors have made in response to the reviews – congratulations on a great paper!

Decision letter (RSPB-2021-2269.R0)

12-Nov-2021

Dear Dr Schulenburg

I am pleased to inform you that your Review manuscript RSPB-2021-2269 entitled "Population size impacts host-pathogen coevolution" has been accepted for publication in Proceedings B.

The referees do not recommend any further changes. Therefore, please proof-read your manuscript carefully and upload your final files for publication. Because the schedule for publication is very tight, it is a condition of publication that you submit the revised version of your manuscript within 7 days. If you do not think you will be able to meet this date please let me know immediately.

To upload your manuscript, log into <http://mc.manuscriptcentral.com/prsb> and enter your Author Centre, where you will find your manuscript title listed under "Manuscripts with Decisions." Under "Actions," click on "Create a Revision." Your manuscript number has been appended to denote a revision.

You will be unable to make your revisions on the originally submitted version of the manuscript. Instead, upload a new version through your Author Centre.

1) A text file of the manuscript (doc, txt, rtf or tex), including the references, tables (including captions) and figure captions. Please remove any tracked changes from the text before submission. PDF files are not an accepted format for the "Main Document".

2) A separate electronic file of each figure (tiff, EPS or print-quality PDF preferred). The format should be produced directly from original creation package, or original software format. Please note that PowerPoint files are not accepted.

3) Electronic supplementary material: this should be contained in a separate file from the main text and the file name should contain the author's name and journal name, e.g. `authorname_procb_ESM_figures.pdf`

All supplementary materials accompanying an accepted article will be treated as in their final form. They will be published alongside the paper on the journal website and posted on the online figshare repository. Files on figshare will be made available approximately one week before the accompanying article so that the supplementary material can be attributed a unique DOI. Please see: <https://royalsociety.org/journals/authors/author-guidelines/>

4) Data-Sharing and data citation

It is a condition of publication that data supporting your paper are made available. Data should be made available either in the electronic supplementary material or through an appropriate repository. Details of how to access data should be included in your paper. Please see <https://royalsociety.org/journals/ethics-policies/data-sharing-mining/> for more details.

<http://datadryad.org/submit?journalID=RSPB&manu=RSPB-2021-2269> which will take you to your unique entry in the Dryad repository.

Once again, thank you for submitting your manuscript to Proceedings B and I look forward to receiving your final version. If you have any questions at all, please do not hesitate to get in touch.

Sincerely,

Professor Hans Heesterbeek

Associate Editor

Comments to Author:

The authors have done a good job revising their ms according to the reviewers' suggestions; no further issues were raised and I also don't have any additional comments.

Reviewer(s)' Comments to Author:

Referee: 2

Comments to the Author(s).

I am happy with the changes the authors have made in response to the reviews – congratulations on a great paper!

Referee: 1

Comments to the Author(s).

none

Sincerely,

Proceedings B

Associate Editor,

Board Member

Comments to Author:

The authors have done a good job revising their ms according to the reviewers' suggestions; no further issues were raised and I also don't have any additional comments.

Reviewer(s)' Comments to Author:

Referee: 2

Comments to the Author(s).

I am happy with the changes the authors have made in response to the reviews – congratulations on a great paper!

Referee: 1

Comments to the Author(s).

none

Decision letter (RSPB-2021-2269.R1)

18-Nov-2021

Dear Dr Schulenburg

I am pleased to inform you that your manuscript entitled "Population size impacts host-pathogen coevolution" has been accepted for publication in Proceedings B.

Data Accessibility section

Open Access

Paper charges

Sincerely,

Appendix A

Dear Sir or Madam,

We thank the associate editor and the two reviewers for the careful evaluation of our manuscript. The comments and suggestions were very valuable. We used them to improve our manuscript, as explained in detail below. We hope that our manuscript is now acceptable for publication.

Please let us know if there are any specific questions!

Best regards,
Andrei Papkou, Hinrich Schulenburg

Detailed response to reviewers' comments

Please note that reviewers' comments are given in gray and italics, while our response is given in blue. We also provide an additional files for the manuscript text, in which changes are tracked and **highlighted in color**. All below line numbers refer to this additional file.

Associate Editor

Board Member: 1

Comments to Author:

In this paper the authors use their C. elegans-B. thuringiensis system to investigate the impact of host population size on host-pathogen coevolutionary dynamics. I think this is an important topic that has received some limited theoretical attention in recent years but is still underexplored. Both reviewers found the paper interesting, but they also identified some shortcomings and made a number of insightful recommendations on how the paper can be improved.

Reviewer 1 raises important points about the way the initial populations were generated. I agree with the reviewer that this is a subtle but potentially important feature of the experimental design that needs to be clearly spelled out and discussed. How would the authors expect a different procedure for setting up the small populations to affect the results? (E.g. with a burn-in without any parasites to reduce genetic diversity to some level closer to what is expected in populations of this size.)

Our reply: Many thanks for this important point. We addressed this comment in our response to Reviewer 1

I also largely agree with reviewer 1's comment about the Discussion - it reads a bit like a cover letter and would benefit from a much more in-depth discussion of the results of the experiment, including the issue raised by Reviewer 2.

Our reply: In response to this comment, we have now substantially re-written part of the Discussion and have taken into account the valuable suggestions of the reviewer.

Finally, I agree with Reviewer 2 that the methods need to be described more exhaustively. Their suggestion to move the Methods section in front of the Results section would help reducing redundancies and is in line with the usual order of sections in Proc B papers.

Our reply: As suggested, we have now moved the Methods section in front of the Result section, in order to enhance clarity and also to reduce redundancies. We further extended parts of the Method section, at least as far as possible under the rather strict Word limit of Proc B. In this context, please note that the very detailed description of methods for the evolution experiment is available in the supplement of our previous publication (as cited in the manuscript), it takes up 20 pages and would thus be too comprehensive for the main text of a Proc B paper. We still hope that the introduced changes help to clarify specific aspects of the research approach used. If not, then we would be grateful for advice from the associate editor as to how more details can be integrated under the standard strict Proc B word limit. Otherwise, see our detailed response to Reviewer 2.

We would like to note that we struggled with our attempt to incorporate the reviewers' suggestions in consideration of the Proc B word limit. To allow the addition of the suggested arguments and details, we thus decided to move panels b and c from Figure 2 and panel b from Figure 3 to the supplement. These panels are not essential to understand the main findings of our study, while still being important for the detailed appreciation of our study's results.

Reviewer(s)' Comments to Author:

Referee: 1

Comments to the Author(s)

This manuscript reports on the outcome of an experimental evolution study with the main treatment factor being population of small and big size. The study is well conducted and mostly clear to follow. The study design is related to that of earlier publications of the group, and is a success story of the lab. While I have no problems with the experiment, its analysis and presentation, but I have problems with the interpretation of the study design and therefore the interpretation of the results and the generalisations made.

The authors are surprised that "... small populations were still able to respond to selection, even if they comprised no more than 100 individuals.". I am not surprised! 23 generations is not a lot. A pop. size of 100 on the other hand is rather big. Given that the response to selection here is only attributable to standing genetic variation (mutations would not play a role with this exp. design), selection simply pulled out the fitter variants that were already at a reasonable frequency.

Our reply: Many thanks for this comment. We agree that high genetic diversity could have allowed for an evolutionary response in the populations consisting of 100 individuals. We now more carefully phrased our description and interpretation of the data in the discussion. We now note that the host responded to selection under both host population sizes, most likely because initial genetic diversity was large under both conditions. The evolutionary response is stronger and significant under the large

population size, at least indicating that genetic drift constrained the response to selection in the small populations. See the second and especially third paragraph of the revised discussion in lines 486-521.

Furthermore, the study suffers from an important misunderstanding of population genetics, which limits what can be concluded from simple theory. A consequence of drift is that small populations have lower genetic diversity than big populations. If a sub-population is created from a big population, it will have initially a too high genetic diversity. Only after several generations burn-in, a new equilibrium will be reached (This is why population geneticists use genetic diversity to estimate (effective) population size). In the current experiment, all populations had been recruited from a big stock. Thus, the genetic diversity, was over proportional high in the small populations. In fact, it was the same in all treatments! Thus, a relatively strong response to selection based on the standing genetic diversity is expected. While I do not consider this fatal for the article, this needs to be discussed and the generalisations made need to be adjusted accordingly. It also needs to be part of the discussion. This has also implications for the bottlenecked populations.

Our reply: Many thanks for this important point. As suggested, we now explain more clearly that both small and large populations initially had similarly high levels of genetic diversity and that this could have determined the response to selection in similar ways under the two imposed population sizes. We also explain that our objective was to specifically assess the influence of differences in census population size in the host and therefore, it was important to keep all else constant (including initial genetic diversity). We then further emphasize that in spite of these initially high levels of genetic diversity, the small host populations responded less strongly to selection, at least indicating that genetic drift influences the evolutionary response more strongly in the small populations. These aspects are now more carefully outlined in the discussion. See the second and third paragraphs of the revised discussion in lines 486-521.

Please note that we did pre-adapt the host population to the general conditions of the experiment, although under only large population size. We decided against pre-adaptation of the host under different population sizes, because this would have created different starting conditions for the distinct treatments and thus it would have required a larger number of control treatments for the main evolution experiment, which would have become difficult – if not impossible – to implement. In such a case, it would have become necessary to expose the populations pre-adapted to small population size to all different population size treatments of the main evolution experiment and also to expose the populations pre-adapted to large population size to all different population size treatment, in order to ensure a full factorial design.

Another issue related to the founding population is the way the founding population was produced. The authors state that: “After the last transfer, all 16 populations were combined, mixed with DMSO (final concentration 15% v/v) and frozen at -80 °C. The pre-adapted population was designated as the ancestral population and used as a starting material for coevolution.” As a consequence of this one-time mixing procedure, the populations exhibit strong linkage disequilibrium (LD), which will decline with time (recombination events) since mixing. LD can have profound impact on the response to selection, in particular when population size is small and the traits under selection being polygenic. Again, this needs

to be worked out and discussed.

Our reply: We agree that the mixing procedure could create LD which then need to be resolved by recombination. The magnitude of LD would depend on how genetically divergent the populations were before mixing. For example, the mixing of two very genetically distinct populations would produce LD of large effect, and mixing of recently separated populations is less likely to generate LD of large magnitude. In our study, all populations used for pre-adaptation were established from the same source population. Considering the large population size, the same selection conditions, the absence of strong directional selection (except of selection imposed by the experimental evolution protocol) and relatively short evolutionary time, we reasoned that these populations had not had enough time to diverge substantially and thus to cause LD of large magnitude after pooling. Nevertheless, LD might be a confounding factor in our experiment when comparing small and large populations as correctly suggested by Reviewer 1. In other words, longer persistence of LD in the small populations could have contributed to the less pronounced coevolutionary response under these conditions. We have incorporated this point into Discussion of the revised manuscript. See lines 491-495 and 522-534.

The discussion is rather boring und hardly informative. After the summary in the first paragraph, we learn across the next two paragraphs why the current study is great and other studies had different types of problems. The following paragraphs are also rather slim in discussing the biology behind the study. I would rather like to read a discussion telling me about the biology of the results presented here, about the possible roles of mutation and standing genetic variation, about the role of the specific experimental design for the outcomes, about the different forms of coevolution that are often discussed, about the role of selfing versus outcrossing of the host, about the long term perspective (what would we expect if the experiment would be run for 1000 generations) and more Thus, I would not add to the discussion, I would replace the current discussion with a new one.

Our reply: We are grateful to Reviewer 1 for encouraging a more informative writing style. We politely disagree that it is uninformative to relate our results to previous work based on similar experimental designs. In fact, such comparisons are essential to place our findings into the context of what has been inferred from previous experimental work on the topic. Nevertheless, as suggested by the reviewer, we now re-focused our discussion and re-wrote most of it. In particular, we now discuss that our results on the influence of host population size is likely due to evolutionary processes rather than epidemiological feedbacks (2nd paragraph of discussion), that host population size influence coevolutionary dynamics irrespective of the high initial level of genetic diversity (3rd paragraph), that recombination and sexual reproduction can generally influence coevolutionary dynamics (4th paragraph), and that host population size can have indirect effects on coevolution (final paragraph). Please note that we decided against discussing all of the points listed by the reviewer, especially the long-term perspective, for which there is no data available and for which the discussion would have become too speculative. See our changes in lines 486-549.

I suggest to make clear that this is an entirely artificial system, where neither the host lives in its typical habitat nor is the pathogen a natural pathogen of these worms. Still, it is a good model system for basic questions.

Our reply: In the revised manuscript, we now explain that we use an established, laboratory-based,

experimental model interaction which serves to study the dynamics of host-parasite interactions under controlled conditions. See changes in lines 81-82 and also the following lines.

The analysis of the experiment is very complex.

(although, I would like if the material would be presented in a less complex form. Not everything presented is needed.)

Our reply: We find it difficult to respond to this comment. In the previous version of the manuscript we put a lot of effort to make results accessible for readers. We included an overview of experimental design in Figure 1 and moved literally all statistical analysis to the supplement. By contrast, the second reviewer would like to see more statistical results in the main text. Reviewer 1 points out that ‘not everything presented is needed’ but does not specify what type of analysis or results are redundant. Nevertheless, we carefully went through the description of our results and simplified the text and the presentation of results, whenever possible and whenever not in contradiction to the requests of reviewer 2. Moreover, we also moved part of the results shown previously in Figures 2 and 3 (all relating to results for host resistance and pathogen virulence, which were not directly under selection in our experiment to the supplement). This may help to keep the results section more focused and also to stick to the word limit of Proc. B. See the revised results section in lines 268-396.

I suggest to use a more standardized design for the figures, e.g. use the same error bars (currently it is standard deviations and 95% C.I.).

Our reply: We have changed the error bars on all figures to standard errors. The only exception are panels in figure 3 showing estimated model coefficients for which we prefer to use 95% confidence intervals.

Referee: 2

Comments to the Author(s)

*This is a very interesting and elegant study investigating the effect of host population size on antagonistic coevolution between *C. elegans* and *B. thuringiensis*. I greatly appreciate the amount of work involved in collecting such an impressive data set!*

My main concern is with the presentation of the work. For example, the authors refer to another paper to explain the evolution experiment. I would prefer a brief summary of what has been done in the main text.

Our reply: We now moved the methods section in front of the results section, in order to make it easier for the reader to understand the design of the work and the research approach used. We now also provide additional details on the design of the evolution experiment, rather than just referring to the previous publication. Please note that we previously already provided an illustration of the design of the evolution experiment and the subsequent phenotypic characterizations in Figure 1, in order to help the reader to understand our approach. This figure is now also quoted in the respective parts of the

introduction (last paragraph), the methods and the Results sections. See the new position of the Methods section and also the additions in lines 127-265.

Second, because of the scale of the study, a lot of information is hidden in the supplementary material. I found it quite hard to follow and interpret the main findings. The majority of statistical analyses (which are all sound) are described in the footnotes in the supplementary tables. Personally, I prefer a statistical summary of the main effects/interactions in the main body of the text, which might provide the reader with a bit more guidance on what results are discussed when.

Our reply: We have moved summary of statistical results for key findings to the main text as suggested by the reviewer (see lines 276-277, 281, 307, 312, 328-332, 356-357, 360-361, and 370). Because of space constraints imposed by Proc B, this could not be done for all statistical analyses. We thus find it important to make all statistical results accessible through the supplement (See supplementary tables of the supplement, which are also cited in the appropriate position in the main text).

Finally, I believe the authors need to be more explicit about the novelty of their work – do population bottlenecks and small sizes in theory differentially affect coevolutionary dynamics? The effect of host population bottlenecks on coevolution has been tested experimentally in bacteria-phage systems. From the introduction it is not immediately clear what this study might add to our understanding of how population size affects coevolutionary dynamics.

Our reply: Many thanks for this advice. We agree that it is important to place our findings into the context of previously published related work. In addition to the studies, which we already cited in the previous version of the introduction (e.g., those based on the Tribolium-beetle model), we now also refer to the study suggested by the reviewer (it was mentioned in the Discussion section of the original manuscript). We further refer to a very recently published study which used a related *C. elegans*-pathogen model (but including a different pathogen taxon). We also maintained our comparison to previous work at the beginning of the discussion, in order to inform the reader about the novel insights obtained from our work. See changes in lines 60-79, 476-485, and 495-505.

Abstract

Please clarify that population size of the host was manipulated and not that of the pathogen.

Our reply: As suggested, we have now further emphasized in the abstract that only host population size was manipulated. See lines 27,28, and 32.

Introduction

Lines 53-62: Can the authors please explain how the effect of host population bottlenecks on antagonistic coevolution is expected to differ from that of small population size? This would clarify the novelty of this study (i.e. the effect of population bottlenecks on coevolution has been investigated experimentally, largely in bacteria-phage systems (e.g. Common & Westra 2019 and Hesse & Buckling

2016). *How does this work add to our understanding?*

Our reply: Many thanks for this important comment and for pointing us to the reference of Common & Westra 2019 (which we missed). We have now incorporated it in our introduction and discussion. In short, these studies point to the importance of feedback between epidemiological/ecological and evolutionary processes for coevolution. Our study is different in that we did not allow for such a feedback in our system and therefore the effect of population size and bottleneck would influence coevolution only by affecting evolutionary processes. We further point out that the consequences of periodic bottlenecks, although common and thus important in nature, are not really clear; they could have similarly strong genetic drift effects than those seen for small populations, or they may be negligible, thus resulting in similar coevolutionary dynamics than those seen for large populations. See changes in lines 60-79 and 545-549.

Lines 75-77: Could the authors add a bit of background on the biology of the system. Does B. thuringiensis impose strong selection on C. elegans? Does it result in death or mainly reduced fecundity? Is B. thuringiensis ahead in the arms race (or is frequency dependent selection driving coevolutionary dynamics) and how is this likely to affect the host when its population sizes are reduced? I would like to know a bit more about this cool model system and why host population size could be a key determinant in driving coevolutionary dynamics.

Our reply: As suggested, we now provide additional information on the interaction of *C. elegans* and *B. thuringiensis* in the introduction, including a concise summary on previous host-pathogen coevolution experiments with this system. We apologize that we could not provide more information due to the Word limit of Proc. B. See changes in lines 82-88.

Results

In general, there are no statistics provided in the result section. While I understand that not all contrasts can be given in the main text (as there are many), personally I would like to see data on significance of main treatment effects (and their interactions) without having to flick back and forth between the main text and the supplementary material.

Our reply: As suggested, we have now added statistical information for key findings to the main text. As noted above, we did not move all statistical results to the main text because of the word limit of Proc B. See changes in lines 276-277, 281, 307, 312, 328-332, 356-357, 360-361, and 370.

Lines 117-122: What I find perhaps more surprising is that adapted and coevolved host lines do not differ in mean egg number across all population sizes tested. This is not mentioned at all in results or in the discussion.

Our reply: We now address this result in the Results and the Discussion section (lines 285-287 and 535-543). See also our more detailed reply below.

Lines 126-127: It is not clear from method section that 8 replicates were tested initially (it only says so in figure legend). Please clarify this, and how changing replicate numbers improved power of statistical models.

Our reply: Many thanks for this suggestion. We now mention explicitly in the main text (and not only in the figure caption) that data shown in Figure 2 includes only 8 replicates per time

point/treatment/population size. In general, including more replicates increases statistical power. But the power function depends on model specification and the exact test used. Those are not directly comparable between the two statistical models. In the first case, we measured change in fecundity across different time points and all different treatments. In the latter case, we only compared different population sizes within the coevolution regimen and at two infection doses. Please also note that detailed phenotyping of all populations had not been feasible. Therefore, for the general comparison of treatments shown in Fig. 2, we focused on a randomly chosen subset of populations, which we now specify in the methods and the results section. See changes in lines 205 following. Similarly, we only used 12 out of 16 replicates to measure the evolution of pathogen virulence at multiple time-points. In all other cases, we used all replicates from the evolution experiment.

Line 129: There were no differences between small and bottlenecked populations either. Please mention this.

Our reply: We did not find a significant difference between small and bottlenecked population. We now mention this aspect in the Results. See changes in lines 285-287. See also our more detailed reply on this topic below.

Lines 135: This is why I would like to see some more detail about the experimental procedure. It is not clear from the method section that eggs from all nematodes were collected and transferred and how this affected survival of bacteria (I presume the authors used this method because it is difficult to distinguish live/dead nematodes?). Is it known whether egg number varies as a function of nematode health (did dead individuals contribute less to future generations in your set up?)

Our reply: We now describe the experimental procedure in more detail, specifically that eggs were collected from all nematodes (dead and alive). Sorting them would have been an impossible task due to the size of the experiment (only large populations had $16 \times 3 \times 3000 = 144000$ nematodes during each transfer). Instead we used mass selection that favored higher egg production upon the exposure to the pathogen. Egg numbers are usually assumed to correlate positively with nematode health, yet we did not measure this specifically for our set-up. However, we do know that under our experimental conditions, almost all nematodes were infected by 40 h and the spore dose used caused high host mortality (mean \pm s.d.: 0.819 ± 0.104 for the ancestral host and pathogen). The more detailed information on the experimental procedure is now provided in lines 172-182. See also further additions to the methods section in lines 172-265.

Lines 138-140: This is not surprising given that main effect of population size was not significant (Table S6). I would omit this table. From my understanding, it is recommended to carry out post hoc comparisons only if main effect/interaction is significant.

Our reply: We agree with the reviewer and removed Table S6

Lines 141-146: This table provides information on contrasts, but there are no statistics on main effects of linear mixed models.

Our reply: To test the differences in pathogen relative fitness at the end-point, we used the non-parametric Wilcoxon test (shown in Table S9). Regression analysis was not used for this dataset. We added this information to the methods and the results (see lines 248-250 and 312; see also a more detailed explanation of the statistical analysis performed in the revised methods section in lines 234-265).

Line 274: should be “middle panel”.

Our reply: We corrected this mistake in original line 174 (see line 348 of the revised manuscript)

Lines 202-206: I think this information should be provided in the introduction when discussing coevolutionary dynamics. Is effect of host population size expected to be different for these different types of dynamics?

Our reply: This is a very interesting point. For our model interaction and our experimental conditions, we previously demonstrated that coevolution leads to aFDS. Therefore, we can here only assess whether variation in population size affects aFDS dynamics across replicates or not. It would clearly be interesting to find out whether population size differentially affects aFDS or recurrent selective sweeps. However, our evolution experiment did not allow us to test this. Thus, a more detailed discussion of this topic would become rather speculative and would go beyond the scope of our manuscript and also beyond the Word limit for Proc B.

Discussion

I miss a discussion on why adapted and coevolved host populations did not differ in their final egg numbers (transfer 23) and how this links to observed FDS in large and bottlenecked populations (although less so).

Our reply: Many thanks for this comment. In response to the reviewer’s comment (and related comments above), we now tried to describe more clearly that some time shift experiments aimed at understanding general responses to selection by exposing hosts or pathogens from different evolution treatments (coevolution, adaptation, control) to their respective ancestral antagonist (illustrated in Figure 1b top panel), while other time-shift experiments aimed at a characterization of the dynamics of change under coevolution conditions only and for which coevolving antagonists from the same replicate were exposed to each other (illustrated in Figure 1b middle and bottom panel). This aspect is now highlighted at the end of the introduction, it is recapitulated in the Methods and Results section, and it is also taken up in the concise summary in the first paragraph of the discussion. See changes in lines 94-99, 205-233, 269-271, 308-309, 333-349, 374-377, and 477-482. Moreover, we now more clearly specify in the Results section (lines 285-287), that there are no significant differences among hosts from coevolution and host-adaptation conditions, when exposed to the ancestral antagonist. Moreover, we now also discuss this finding in the last paragraph of the discussion (lines 535-543).

As now explained in more detail, the fitness data for hosts from the coevolution and the host-adaptation treatments were obtained by infecting the nematodes with ancestral pathogen (Figure 2a). By using the same ancestral antagonist for all, this allowed us to compare the different evolution treatments (coevolution, adaptation, control) to each other. However, this comparison can only reveal general effects, but not necessarily those most relevant for the coevolutionary interaction, which may depend on presence of the co-adapting antagonist. This is the reason why we performed the additional time-shift experiments, however in these cases only for the coevolution treatments, where we then exposed the coevolved hosts to their coadapted pathogens from different time-points and always from the same biological replicate. The results are shown in Figures 3 and 4. In figure 3, we can see that egg number differs depending on the pathogen used in the assay (i.e. black lines vs orange and red lines). Moreover, there are no GxG interactions for the worms from generation 23 (large population size, Figure 3b) while such GxG interactions are found for the worms from generation 10. In other words, the absence of

differences between hosts from coevolution versus host-adaptation treatments is likely explained by the fact that the hosts are all exposed to the ancestral pathogen. By contrast, time-shift experiments between the coevolved antagonists are likely the most relevant comparison, they are constrained to the coevolution treatment only, and they do indeed demonstrate a significant effect of population size on coevolution.

Methods

Lines 305-327: The authors cite a related paper (Papkou et al 2019) to explain the experimental design. As a result, it is quite hard to understand the experimental evolution experiment (How do transfers work? How do you separate partners in evolution/adaptation treatment? How were population sizes remained constant within a season? (hosts have a generation time of one week)? I would like to see a brief summary of the actual experiment. I do realise that space is limited but perhaps this could be solved by placing the method section directly after the introduction. This way, the method summary at the beginning of the result section is superfluous and can be removed to free up some space.

Our reply: Many thanks. We agree and tried to integrate a more detailed description of the methods of the evolution experiment into the methods section. At least as far as the word limit of Proc B permits. Please note that the very detailed description of the evolution-experiment-methods covers 20 pages and would thus be too comprehensive for the main text of a Proc B paper. Nevertheless, we now provide a more detailed overview of the approaches used. The very exact details of the methods then still remain accessible for the interested reader through our previous paper. In addition, we now also placed the methods section before the results section, as suggested, allowing us to delete some of the methods description which were previously given in the results section. See changes in lines 127 and following.

Line 341: Are the exact sample sizes given somewhere (or are they always 30 per population)? How much variation is there within a population? By averaging, this information is lost (could include this in lme?).

Our reply: We added exact sample size for each measurement into Supplementary data file. For host fecundity, the mean and median of all sample sizes were close to 30 with only a few outliers, therefore we decided not to use weights in corresponding models.

Line 338: It is not immediately clear here whether these are ancestral spores or whether this is a general procedure used for all different host-parasite combinations. I like the brief summary given in lines 110-116 (including reference to panel 1b), which could be placed here to clarify this?

Our reply: Many thanks for this comment. For this assay we used the spores from the ancestral pathogen. We now specify this detail in this context and also refer to Figure 1b, in order to enhance clarity of the approach used (see lines 205-210). Please note that the Methods have now been moved before the Results section, making it easier for the reader to follow the description of results. See changes in lines 127 and following.

Line 349: Please provide formula for relative fitness (it would seem from graph it is actually the selection coefficient).

Our reply: As suggested, we now provide details as to how relative fitness of competitors was calculated. Because we do not know the actual growth dynamics upon infection for the pathogen, we cannot reliably determine generation time required for re-scaling of the selection coefficient. Therefore,

the changes in proportion of pathogen genotypes during competition are per one infection round and not per one cell division (i.e. not directly interpreted as selection coefficient). See changes in lines 211-220.

Lines 361-362: Please specify the structure of the random effect as well as the explanatory variables and link function used (I presume a binomial error structure was used in case of survival data, but this is not provided in main text?). How were different model fits compared? Where assumptions of models met/tested (survival data are often skewed/zero-inflated)? This is not clear at all from statistics description and only becomes apparent from footnotes in the supplementary tables.

Our reply:

Random effects were modeled as random intercept for each biological replicate (i.e. +(1|POPID) following lme4 package specification). The binomial model did not produce a good fit. Therefore, we have used boxcox transformations of the response variable (proportion of dead nematodes). This information is now included in the Methods section (see lines 234 and following). Please see below, the comparison between the model using the binomial distribution and the model using boxcox transformation (lambda=2).

To account for non-linearity between response variable and time we used a polynomial model, which provided a significantly better fit as determined by model diagnostics (likelihood ratio test, QQ plot, fitted vs. residuals and plotting the model prediction over the actual data). Given complexity of this dataset and the “danger” of overfitting for polynomial models, we were very careful with interpreting the results of this particular analysis. We now provide a more detailed description of our statistical analysis in the revised manuscript (see lines 234 and following). Regarding the problem of zero-inflation, we did not have it as there were very few data points that have 100% survival or 100% mortality.

Lines 368-369: I presume post hoc comparisons were only performed when main effects/interactions were significant (see above for model comparison)?

Our reply: Reviewer 2 is correct that post-hoc comparisons do not make sense if the main effects are not significant we therefore leave out the former Table S13 (the main effects are not significant in table S12)

Figure 1: "anatagoist" in panel B should be "antagonist"

Our reply: We corrected this typo.